# Chronic Subdural Hematomas—A Retrospective Analysis of the Internal Architecture and Evaluation of Risk Factors for Recurrences After Surgical Therapy

**DOI:** 10.3390/diagnostics14222494

**Published:** 2024-11-07

**Authors:** Nadja Grübel, Christine Klemptner, Benjamin Mayer, Frank Runck, Gregor Durner, Christian Rainer Wirtz, Andrej Pala

**Affiliations:** 1Department of Neurosurgery, Ulm University, Lindenallee 2, 89312 Günzburg, Germany; nc.gruebel@gmail.com (N.G.);; 2Institution of Epidemiology and Medical Biometry, Ulm University, Schwabstrs 13, 89075 Ulm, Germany; 3Department of Neuroradiology, Ulm University, Lindenallee 2, 89312 Günzburg, Germany

**Keywords:** chronic subdural hematoma, recurrence, internal architecture, outcome, hemispheric type

## Abstract

**Background:** Chronic subdural hematoma (CSDH) is increasingly common due to the aging population and widespread use of anticoagulant and antiplatelet medications. The objective of this study is to examine the internal composition of CSDH and explore potential risk factors associated with its recurrence. **Methods:** This retrospective study analyzed data from 189 patients who underwent surgery in our department between 2014 and 2018. Recorded data included demographics, clinical information, details of surgical interventions, computer tomography (CT) scans, neurological assessments, and follow-up data. The outcome was evaluated clinically and through CT follow-up conducted 4–12 weeks post-surgery. CT scans measured various parameters, including hematoma thickness, hyperdense regions, chronic components, and membrane presence. **Results:** Patients after the evacuation of CSDH were significantly more common males (66.1%, *p* > 0.001) had a significantly higher BMI (*p* < 0.001, 61.6%), arterial hypertension (*p* < 0.001, 68.3%), and the intake of anticoagulant therapy (*p* < 0.001, 58%). The recurrence rate was 18.6% after 4 weeks and 2.1% after 8–12 weeks. After uni- and multivariable analysis, the initial hemispheric type (*p* = 0.019, HR: 3.191; *p* = 0.012, HR: 3.810) and the increasing preoperative midline shift in CT (*p* = 0.028, HR: 1.114; *p* = 0.041, HR: 1.107) were found as independent predictors for recurrence. Overall, outcomes were favorable with a modified Rankin scale (mRS) of 0–2 at discharge (72%), after 4 (89.7%) and 12 (87%) weeks. **Conclusion:** According to our data, increasing midline shift before surgery and initial hemispheric type of hematoma were independent predictors of recurrence. Most patients achieved an excellent outcome with a low-risk profile.

## 1. Introduction

Chronic subdural hematoma (CSDH) is a common disease in elderly (>65 years) patients and a frequent disease pattern in neurosurgical daily practice with a high recurrence rate after surgery [1,2]. The estimated 1-year incidence is 5–58/100 000 [1,2]. Due to the aging population and the more frequent use of anticoagulation and antiplatelet medication in cardio-cerebrovascular disease prevention, the diagnosis of CSDH can be called a public health issue. In Germany, the number of people from the age of 67 is estimated to increase by 22% until the year 2035 [1]. According to these data, the incidence will approximately increase over the next decade [1]. The need for proper treatment concepts is now more than ever.

### 1.1. Risk Factors for CSDH and Recurrent CSDH

Known risk factors for CSDH are patient age (>65 years), male gender, recurrent falls and mild head trauma [3,4,5,6], the use of antithrombotic or anticoagulant therapy [7,8], or other factors such as liver cirrhosis, chronic renal failure, hematologic diseases, and chronic alcohol abuse [9]. Further risk factors can be cardiovascular diseases such as arterial hypertension, diabetes mellitus, and obesity [1], dementia/cerebral atrophy, and coexisting hydrocephalus [10]. According to Rauhala et al., there are various disclosures in the literature regarding recurrence rates from 5 to 30% [1,2,11].

### 1.2. Radiologic Findings and Internal Architecture of CSDH

Different studies have shown large discrepancies in radiological risk factors due to variations in radiological measurement techniques and hematoma classification [12,13]. Subdural hematomas can radiologically be classified using the density and subdivided into hyperdense, isodense, and hypodense hematomas [14]. Nakaguchi et al. classified the CSDH for the first time according to their internal architecture into five different types: homogeneous, laminar, separated, gradation, and trabecular [15]. For this study, we classified the CSDH regarding their division but specified the classification by adding a 6th type, the septate type, in which the hematoma is divided by multiple hyperdense septs. Additionally, we crosschecked the connections between recurrence and internal architecture. Following Nakaguchi et al., we defined the spatial distribution of the CSDH into five types: the fronto-temporo-parietal-base type, the interhemispheric type, the parieto-occipital type, the frontoparietal type, and the hemispheric type. The aim of this study is to investigate the relevance of different risk factors for developing a CSDH or a recurrence depending on different internal architectures. We hypothesized an association between the coexistence of several risk factors in one individual and radiological predictions regarding a recurrent hematoma. The aim is to better understand the disease patterns and, if necessary, adjust regarding therapy or follow-up examination.

## 2. Materials and Methods

### 2.1. Patients and Follow-Up

A monocentric retrospective data-based analysis of 189 patients with CSDH who underwent surgical treatment at the Department of Neurosurgery in Günzburg, University Hospital Ulm, between 01/2014 and 12/2018 was performed. The patient records, treatment documentation, and imaging data were used to analyze the demographic data, patient history, neurological status, and classification of the CSDH by using CT- or magnetic resonance imaging (MRI), surgical treatment and perioperative management, and follow-up examination after 4 to 12 weeks. The diagnosis was based on imaging before hospital admission. The clinical outcome was measured using the modified Rankin scale (mRS). A recurrent hematoma was considered the reaccumulating of blood on the ipsilateral side in conjunction with clinical–neurological symptoms [16]. According to CT imaging findings, different cytoarchitecture types of CSDH were defined as published by Nakaguchi et al. [15]. For this study, we refined the classification by adding a sixth type, the septate type, where the hematoma is divided into numerous hyperdense septa in the CT images. Regarding the evaluation of a midline shift, we differentiated between a continuous midline shift and whether there was a midline shift at all (yes/no). All images were evaluated in cooperation with the department of neuroradiology. The study was approved by the University of Ulm’s ethics committee and registered under reference number 133/19.

### 2.2. Statistics

The statistical analysis was performed by using the statistical package SPSS 25 (IBM Inc., Armonk, NY, USA). For the quantitative evaluation, we assessed parameters like median, mean value, and standard deviation. For the qualitative evaluation, we used absolute and relative frequency calculations. Comparisons were assessed with the Mann–Whitney U test and furthermore, we performed a binary logistic regression analysis of surgical outcome and recurrence. All variables that showed a significant influence in the univariable model were used for multivariable analysis. *p*-values < 0.05 were assumed to be statistically significant. The clinical outcome was dichotomized and defined as a good outcome if mRS achieved 0–2 and a bad outcome 3–4. The potentially influential variables included age, midline shift, anticoagulation therapy, type of hematoma based on cytoarchitecture, sex, atrial hypertension, BMI, duration of hospital stay, complications after surgery, renewed surgery, and comorbidities.

## 3. Results

### 3.1. General Characteristics

This study enrolled a total of 189 patients diagnosed with CSDH. Basic patient demographics and characteristics of CSDH are outlined in Table 1. Radiological evaluation was not feasible in 10 patients (5.3%) due to poor visualization. Only patients who underwent surgical treatment were included in the study. The treated group observed a significant male predominance (Table 1). Initial clinical symptoms leading to diagnosis included paresis in 48.6% (*n* = 90) of 185 patients, aphasia in 33.7% (*n* = 63) of 187 patients, and orientation disorders in 41.6% (*n* = 77) of 185 patients. Additionally, the Glasgow coma scale (GCS) score at admission was assessed, revealing that most patients (92.1%; *n* = 174) had a GCS score exceeding 12 points (*p* < 0.001).

### 3.2. Surgery and Complications

The most frequently performed surgical procedure was burr hole trepanation. It was performed unilaterally as a single burr hole on one side of the skull in 76.2% (*n* = 144) of cases. In contrast, bilateral burr hole trepanation due to subdural hematoma on both sides was necessary in 23.8% (*n* = 45) of cases. Only 3.2% (*n* = 6) of patients underwent a small craniotomy. The average duration of surgery was 30.4 min (SD 15.4; median 27). Subdural drainage was inserted in 93.7% (*n* = 177, *p* < 0.001) cases. A total of 11 patients (5.8%) required revision surgery.

Regarding complications, secondary hemorrhage occurred in 12 patients (6.3%) and wound-healing disorders were observed in three cases (1.6%). Notably, 91.5% (*n* = 173, *p* < 0.001) of patients experienced no postoperative complications at all. Table 2 summarizes the characteristics of surgical therapy and associated complications.

### 3.3. Radiological Evaluation and Internal Architecture at Initial Diagnosis

CT images from 179 patients were examined, revealing bilateral CSDH in 33% (*n* = 59) of cases. The average width of CSDH was measured at 21.7 mm (SD 7.1 mm, range min 4.2 mm-max 45.2 mm). In the initial CT scans, a midline shift was detected in 86.1% (*n* = 155; *p* < 0.001) of 180 patients, with an average diameter of 7.3 mm (SD 4.2 mm; range from 1.3 mm to a max. of 20.4 mm). The spatial distribution of findings is summarized in Table 3. Upon a more detailed analysis of internal structure, our data revealed that the septate type was the most common, accounting for 44% (*n* = 76, *p* < 0.001) of 172 patients (Table 4).

### 3.4. Radiological Evaluation and Internal Architecture of Recurrent Hematomas

In total, 188 out of 189 patients were assessed for recurrent hematoma at 4, 8, and 12 weeks post-surgery. One patient passed away during their hospital stay and was therefore excluded from the recurrent hematoma analysis. Our findings revealed recurrent hematoma in 35 cases (18.6%) after 4 weeks and in four patients (2.1%) after 8–12 weeks. Due to missing data, midline shift assessments were conducted in 32 out of the 35 cases. CT imaging indicated a midline shift in 71.9% of these cases, with an average shift of 5.8 mm (range 1.5–15.7 mm). Additionally, patients with recurrent hematoma after 8–12 weeks exhibited midline shifts in two out of four cases (1.6 mm and 12.6 mm). Spatial distribution analysis was performed in 30 patients with recurrent hematoma, revealing the frontoparietal convex type as the most prevalent in 70% of cases (*n* = 21, *p* < 0.001). Furthermore, in 60% of cases, the recurrent hematoma exhibited septation (*n* = 18; *p* < 0.001). Overall, 39 cases (20.7%) demonstrated recurrent hematoma, with 24 cases (66.7%) showing a change in internal architecture on CT imaging.

### 3.5. Risk Factors and Comorbidities for CSDH and Recurrent CSDH

Body mass index (BMI) was evaluated in 177 patients, with the majority (61.6%; *n* = 109) having a BMI exceeding 25 kg/(m)^2^ (*p* > 0.001). Arterial hypertension was observed in 127 out of 186 patients (68.3%; *p* < 0.001), indicating a significantly common comorbidity. Additionally, 58% (*n* = 109) of 188 patients reported anticoagulation intake (*p* < 0.001), while 108 patients (57.4%) recalled a prior trauma before diagnosis. Binary logistic regression analysis revealed no statistically significant impact of arterial hypertension, BMI, patient age, gender, anticoagulation use, or Glasgow coma scale (GCS) score < 13 as risk factors for developing recurrent hematoma after 4 and 12 weeks. Univariate and multivariate analyses of midline shift data indicated that a larger midline shift was associated with a significantly higher likelihood of recurrent hematoma development after 4 and 12 weeks (Table 5). Specifically, the progression of midline shift influenced the recurrence rate, while the presence of midline shift, operation time, CSDH width, and GCS showed no significant influence on recurrent hematomas. Examining the intracranial alignment of recurrent hematomas, it was notable that the initial hemispheric type increased the likelihood of recurrence after 4 and 12 weeks (Table 5). Other intracranial alignment types and internal architecture did not exhibit a statistically significant influence on the recurrence rate after 4 and 12 weeks.

### 3.6. Outcome

Most patients (*n* = 131, 69%) could return home after surgery. A total of 23 (12.2%) were discharged to a rehabilitation center, 20 (10.6%) were discharged to another hospital, 9 (4.8%) were discharged to a nursing home, 4 (2.1%) were discharged to short-term care, and only 1 patient (0.5%) died during a hospital stay. Neurological outcome was measured using the mRS and Glasgow outcome scale (GOS). Overall, outcomes after surgical therapy were favorable, with an excellent functional outcome, with an mRS of 0–2 at discharge (72%), after 4 (89.7%), and 12 (87%) weeks. After logistic regression, higher age (*p* = 0.05, HR 1.032, Cl 95% 1000–1066), comorbidities (*p* = 0.032, HR 3.225, Cl 95% 1107–9402), and prolonged hospital stay (*p* = 0.003, HR 1.118, Cl 95% 1039–1202) had a statistical influence on the mRS > 2 points. There was no statistically significant connection between the mRS and the internal architecture of CSDH.

## 4. Discussion

We conducted a retrospective analysis of patients who underwent surgical treatment for CSDH, aiming to identify factors influencing outcomes and recurrence rates. Our findings revealed a recurrence rate of 20.7% (*n* = 39). The recurrence rate was 18.6% after 4 weeks and 2.1% after 8–12 weeks, which is relatively low compared to published data, which ranges from 5–30% [2,11]. Moreover, our results indicated a higher likelihood of recurrent hematoma in cases with progressive midline shift and identification of a hemispheric type in preoperative imaging. Based on these findings, we propose considering additional therapies, such as meningeal artery embolization, in select cases to mitigate recurrence [17]. Interestingly, our analysis of internal cytoarchitecture revealed that the septate type was not associated with an increased risk of recurrent hematoma [15]. Despite being the most common type in our cohort, the septate type exhibited the least architectural change (35.7%). In contrast, other types showed architectural changes ranging from 71% to 100%. This observation suggests that the septate type may represent a more advanced stage of CSDH, existing longer than other types and demonstrating minimal changes in the internal architecture.

Consequently, the septate type’s stability may explain why it was not identified as a risk factor for recurrent hematomas. Overall, most of our patients achieved excellent clinical outcomes, with over 90% attaining a Glasgow outcome scale (GOS) score of 4 or 5 points, consistent with findings reported by other authors [18]. This can indeed also be explained by the fact that most patients (92.1%; *n* = 174) had a relatively mild GCS of 12 at the time of initial diagnosis.

### 4.1. Surgical Therapy and Complications

There are numerous treatment options for patients with CSDH. From conservative therapy to burr hole, twist–drill burr hole trepanation, with or without drain placement as well as craniotomies. Burr hole trepanation with insertion of subdural drainage is a commonly used surgery technique for CSDH [17,18]. We mostly used single burr hole trepanation with subdural drain placement. New treatment options, especially for patients with recurrent hematomas or/and anticoagulant therapy intake, are middle meningeal artery (MMA) embolization using particles and coils [19,20,21,22]. Patients with clinical or imaging risk factors for developing a recurrent hematoma could benefit from MMA embolization in addition to surgery [19]. In this study, all in all, the rate of secondary hemorrhage (6.3%; *n* = 12), wound healing disorders (1.6%; *n* = 3), and necessary reoperations (5.8%; *n* = 11) was low. Our results confirm the data from Martinez-Perez et al. [23]. A statistical relationship between the previously stated complications and recurrent hematomas could not be found. Postoperatively, 94.1% of the patients showed no neurological deficit. The data show that patients with surgically treated CSDH have a favorable outcome. These results are supported by an Uno et al. (2017) study. Their data suggest that surgical therapy leads in 70–90% of the cases to quickly improving symptoms [24].

### 4.2. Risk Factors for CSDH and Recurrent CSDH

#### 4.2.1. Demographic Data

Many authors have described the association between advanced age and male gender with the development of CSDH, a correlation that our data also supports [5,21,25]. However, in our study, we found no significant connection between advanced age and recurrent CSDH. Nevertheless, a significant association between advanced age and unfavorable outcomes at discharge was observed (*p* = 0.05). This may be explained by the presence of more comorbidities in older individuals [6,23].

Our findings corroborate the established notion that CSDH is often preceded by minor head trauma (57.4%). Additionally, higher BMI (61.6%) and arterial hypertension (68.3%) were identified as common comorbidities among patients with CSDH. These results are consistent with the existing literature and align with expectations [1,8,9,16,19,26].

#### 4.2.2. Anticoagulation

The literature consistently indicates that the use of anticoagulation medication poses a risk factor for the initial development of CSDH [3,6,8,9,27,28,29]. In recent years, there has been a notable increase in the prescription of anticoagulant therapy. However, the role of these medications in the recurrence of CSDH remains unclear. In our study, 58% of patients were on anticoagulant therapy, a substantially higher figure compared to previous data reported by Gelabert-González in 2005 (12.2%), but more in line with findings from Borger et al. in 2012 (37–52.8%) [30,31]. This indicates a significant rise in anticoagulant therapy usage over time. However, our data revealed no significant correlation between anticoagulant therapy and recurrent CSDH. This finding is consistent with publications by Lindvall et al. [7] but contradicts the findings of Wada et al. [32]. It remains uncertain whether there is a noteworthy distinction between anticoagulation and antiplatelet therapy since both are seldom differentiated in the literature.

Furthermore, in our study, associated complications such as secondary hemorrhage and wound healing disorders were not elevated among patients on anticoagulant therapy. Nevertheless, there are currently conflicting viewpoints in the literature regarding the significance of anticoagulant therapy in the development of CSDH.

#### 4.2.3. Radiological Findings

In 81.6% of our patients, initial CT imaging revealed a midline shift with an average displacement of 7.3 mm. While a midline shift alone did not significantly affect hematoma recurrence, our data revealed a notable correlation between progressive midline shift and recurrent hematoma following uni- and multivariable analysis. Specifically, larger midline shifts were associated with an increased incidence of recurrent hematomas after 4 and 12 weeks, a finding also reported by Kim et al. [33] and Greuter et al. [34] but not observed by Stanisic et al. [35]. Despite the heightened risk of recurrence, we found no significant associations between midline shifts and unfavorable clinical outcomes (mRS > 2). Regarding spatial distribution, the multivariable analysis indicated that the hemispheric type was more frequently associated with recurrent CSDH (*p* = 0.012; *p* = 0.029), significantly influencing recurrent CSDH development. The fronto-parietal-convex type emerged as the most common CSDH subtype, accounting for 44.9% of cases. While it was also the most prevalent type in recurrent hematomas, no statistical association with CSDH recurrence was identified. These findings are noteworthy, considering that most studies do not classify CSDH subtypes, suggesting that such a classification may benefit clinical practice. Based on our results, the categorization of CSDH based on spatial distribution is recommended for daily practice. Notably, a large midline shift and a hemispheric type should prompt clinicians to schedule follow-up appointments for clinical and imaging examinations to detect recurrent hematomas before they become symptomatic. The pathophysiology of CSDH development is subject to debate in the literature. Various hypotheses, including inflammatory reactions due to pseudomembrane formation and angiogenesis, as well as trauma-induced injury to bridging veins, have been proposed. Recent theories combine these ideas, suggesting that CSDH develops after minor head trauma, leading to dura–neurothel injury, the release of Vascular Endothelial Growth Factor (VEGF), and increased capillary vessel growth, potentially causing recurrent microbleeds and local coagulopathy [3,36,37,38]. These findings may explain the diverse internal architecture observed in CT imaging of CSDH, suggesting that the septate type represents the final stage of CSDH.

In contrast to other publications, our study did not find significant associations between internal architecture and recurrent CSDH development. Specifically, the septate type was not linked to a higher recurrence rate. Overall, the septate type was the most prevalent initial and recurrent CSDH subtype in our study. Notably, changes in internal architecture were less common in the septate type compared to other subtypes, further supporting the notion that it represents a later stage of CSDH development. Table 6 summarizes the ‘traditional’ risk factors for CSDH, while Table 7 outlines risk factors for recurrent CSDH alongside references, comparing our findings.

### 4.3. Limitations

The main limitations of this study stem from its retrospective design and the absence of a conservatively treated control group. The treatment approach for chronic subdural hematoma is highly individualized and influenced by various factors. Despite identifying a correlation between larger midline shift and recurrent CSDH, we could not establish a definitive threshold for midline deviation. Further investigations are warranted to determine a threshold for defining an appropriate interval for follow-up appointments. Our findings predominantly rely on patients who underwent surgical treatment involving burr hole trepanation and subdural drainage insertion. As a result, we cannot conclude other surgical techniques, although our results were obtained using a consistent method. Nevertheless, owing to the substantial patient cohort, regular follow-up assessments, and comprehensive CT imaging, this study provides valuable insights into disease patterns associated with recurrent hematoma development and unfavorable outcomes.

## 5. Conclusions

Based on our findings, most patients who underwent surgical treatment for CSDH with a single burr hole and subdural drainage achieved excellent outcomes. Additionally, escalating midline shift pre-surgery, along with the initial hemispheric hematoma type, emerged as independent predictors of hematoma recurrence. In an upcoming prospective registry study, we aim to evaluate the impact of various CSDH treatment modalities and the reduction of modifiable risk factors on recurrence rates and complications in CSDH patients.

## Figures and Tables

**Table 1 diagnostics-14-02494-t001:** Patients’ and CSDH characteristics.

Patient Characteristics	
*n*	189
Age (years, median)	76
Age > 70 years	78.8% (149)
Males	66.1% (125)
Time between trauma and diagnostic (median)	24 days
Unilateral hematoma	67% (120)

**Table 2 diagnostics-14-02494-t002:** Characteristics for surgical therapy and complications.

Type of Surgery	Total % (*n*)
Insertion of a subdural drainage	93.7% (177)
Bilateral burr hole	23.8% (45)
Unilateral burr hole	76.2% (144)
Surgery time (minutes, median)	27
Secondary hemorrhage	93.7% (177)
Wound infection	1.6% (3)

**Table 3 diagnostics-14-02494-t003:** Spatial distribution of CSDH at initial diagnosis.

Spatial Distribution	Total % (178)
Fronto-parietal	44.9% (80)
Fronto-temporo-parietal	40.4% (72)
Hemispheric	78.8% (149)
Parieto-occipital	11.8% (21)
Interhemispheric	0

**Table 4 diagnostics-14-02494-t004:** Internal architecture of CSDH at initial diagnosis.

Internal Architecture	Total % (172)
Septate	44% (76)
Gradiation	18% (31)
Homogeneous	10.5% (18)
Trabecular	10.5% (18)
Separated	9.9% (17)
Laminar	5.8% (10)
Hygroma	1.3% (2)

Definition of the internal architecture of subdural hematoma modified after Nakaguchi et al. [15]: Septate type: The hematoma is divided into numerous hyperdense septa; a uniform density in the CT image characterizes homogeneous type. This density can be hypo-, iso-, or hyperdense. The laminar type consists of a thin, hyperdense layer on the inner surface of the membrane, with the rest of the hematoma being homogeneous. The separate type is defined by two components of different densities. The boundary is visible, separating the upper hypodense portion from the lower hyperdense portion. Gradation type: If there is no clear boundary but rather a blending of both components, it is referred to as a grading type. In this case, the two hematoma portions are mixed. The trabecular type has an inhomogeneous density. Within the hematoma is a higher-density septum than the rest of the hematoma. Hygroma: Liquor intense subdural fluid collection.

**Table 5 diagnostics-14-02494-t005:** Univariate and multivariable analyses of midline shift and spatial distribution after 4 and 12 weeks.

Variables	Univariate Analysis			Multivariable Analysis		
	HR	95 % CI	*p*	HR	95 %	*p*
Midline shift (continuous), 4 weeks	1.114	1.012–1.226	0.028	1.107	1.004–1.221	0.041
Midline shift (continuous), 12 weeks	1.117	1.017–1.226	0.020	1.112	1.011–1.224	0.029
Hemispheric type, 4 weeks	3.191	1.207–8.437	0.019	3.810	1.345–10.790	0.012
Hemispheric type, 12 weeks	2.950	1.121–7.760	0.028	3.650	1.292–10.310	0.015

**Table 6 diagnostics-14-02494-t006:** Comparison of risk factors in the literature vs. our data for CSDH.

Risk Factors and Diseases	References	Our Data
Age (>65 years)	[1,5,10,22,23]	yes
Male gender	[5,6,22,23]	yes
Mild head trauma	[1,3,8,9,16,17]	yes
Cardiovascular parameters		
Systemic hypertension	[1,8,9,16,17]	yes
Increased BMI	[1,8,9,16,17]	yes
Drug-related use of antiplatelet/anticoagulant therapy	[3,5,7,8,21,22,23]	yes
Dementia/cerebral atrophy	[1]	unlevied
Acoholism	[9,10]	unlevied
Coexisting hydrocephalus	[6]	unlevied
Radiological parameters		
Bilateral cSDH	[12]	no
Hematoma thickness and midline shift	[12]	yes
Hematoma density and internal architecture	[12]	no
Hematoma volume	[12]	no

**Table 7 diagnostics-14-02494-t007:** Comparison of risk factors in the literature vs. our data for recurrent CSDH.

Risk Factors	References	Our Data
Surgical parameters		
No subdural drainage	[13,26]	no
Radiological parameters		
Hematoma density	[27,28]	no
Hematoma thickness	[13]	no
Midline shift	[29,30]	yes
Bilateralism	[28]	no
Separate/laminar type	[31,32]	no
Hemispheric type	[12]	yes
Age	[22,33,34]	no
Gender	[22,33,34]	no
Drug-related use of antithrombotic or anticoagulant therapy	[35]	no

## Data Availability

The authors confirm that the data supporting this study’s findings are available within the article. Raw data were generated in our department. Derived data supporting this study’s findings are available from the corresponding author on request.

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
