# Peer review of "Chronic Subdural Hematomas—A Retrospective Analysis of the Internal Architecture and Evaluation of Risk Factors for Recurrences After Surgical Therapy"

_diagnostics, 2024, doi:10.3390/diagnostics14222494_

Round 1

Reviewer 1 Report

Comments and Suggestions for Authors

The reviewer is neurosurgeon end  chSDH is  one  of his favorable topics.

This article   can be  signed   as  good  job  on old theme  ( „ in country of rev. „ old west”…)

189  pts , burr hole  with drainage  was  as preferred  typ of surgery.

According CT  was No of  recurrency 18,6% after 4 weeks and   after 12 weeks only 2,1 %

Only 3.2% (n=6) of patients underwent a small craniotomy

A total of 11 patients (5.8%)  had complication with repeated surgery – it means acute revision for bleeding?

In CONCLUSIN : only  midline shift in beginning was   significant factor a hemispheric type of SDH supporting recurrency, surprisingly   not  internal  architecture of SDH according Nagakuchi .

Any   else  factor , usually  followed- up (   air  in   residual cavity,  quality of draining fluid,  type of drainage , perioperative  brain tendency to    reexpansion , quality of membrane….etc.)

Average  time  interval  for  healing of SDH after surgery   is about 7-10 weeks.

Definicy of   true recidivans chSDH is problematic, if authors have after 12  weeks only 2,1% pts , symptomatic and   needs repeated surgery or embolization, so   we  can only congratulate  to excellent  results, because    average  No in many studies  about 10%.!

 It have  be clearly  explained.

Author Response

Thank you for your detailed feedback and observations. We agree that the recurrence rate of 2.1% at 12 weeks is notably low compared to the average rates reported in the literature, and we will ensure to clarify this in the manuscript (pt 184-186, page 5/10). Additionally, we acknowledge the significance of initial midline shift as a predictor of recurrence and will emphasize why internal hematoma architecture did not play a significant role, according to our findings. We appreciate your recognition of the results and your insightful comments on the study.

In 11 patients, a revision was performed, including cases of acute bleeding and insufficient decompression. The timing of these revisions is not specified. Thus, the total number of revisions is 5.8%.

Reviewer 2 Report

Comments and Suggestions for Authors

The authors present a manuscript describing internal architecture and risk factors for reoccurrence of a subdural hematoma following surgical therapy.  Overall, the manuscript well-conceived and written.  Some suggestions below to improve clarity of the information provided:

Abstract- It would be helpful to define the term mRS 

Materials and Methods:

-page 2 line 82: more details are needed to describe the term "septate type".

Results:

-page 3 line 108:  It appears that the majority of the patients in the sample were identified as "Mild " define by GPS. Can you provide further insights on what this means in the results or discussion.

-page 3 line 113: "burr hole trepanation"-please provide a brief description and differentiation between unilateral and bilateral.

-page 4 line 131: It would be helpful to have a definition of the internal architecture described in Table 4. This can be done via a line under the table.

References:

-Reference #16 needs a year it was published.

Author Response

Thank you for your thoughtful review of our manuscript regarding the internal architecture and risk factors for subdural hematoma recurrence following surgical therapy. We appreciate your positive feedback and suggestions for improvement. Here are our responses to your comments:

Abstract: We included a definition of the modified Rankin Scale (mRS) on page 1, line 26.

Materials and Methods: Regarding the term "septate type," we have specified that this refers to hematomas divided into numerous hyperdense septa in CT imaging (page 2, lines 82-83).

Results: For most patients identified as "Mild" according to the Glasgow Coma Scale (GCS), further insights into this classification are discussed on page 6, lines 201-203.

We have provided a brief description and differentiation between unilateral and bilateral burr hole trepanation on page 3, lines 114-117.

Internal Architecture: Our manuscript now includes a definition of the internal architecture of subdural hematomas, modified after Nakaguchi et al., in Table 4.

Septate type: hematoma divided into numerous hyperdense septa.

Homogeneous type: characterized by uniform density, which can be hypo-, iso-, or hyperdense.

Laminar type: featuring a thin hyperdense layer on the inner surface of the membrane, with the rest of the hematoma being homogeneous.

Separate type: defined by two components of different densities, with a visible boundary separating the upper hypodense portion from the lower hyperdense portion.

Gradation type: where there is a blending of both components without a clear boundary.

Trabecular type: exhibiting inhomogeneous density, with a higher-density septum within the hematoma.

Hygroma refers to a liquor-intense subdural fluid collection.

References: We ensured that reference #16 includes the year of publication.

All changes are marked yellow.

Thank you once again for your valuable insights, which will enhance the clarity of our manuscript.